

# A Climate Data Record of Stratospheric Aerosols

Viktoria F. Sofieva[1], Alexei Rozanov[2], Monika Szelag[1], John P. Burrows[2], Christian Retscher[3], Robert Damadeo[4], Doug Degenstein[5], Landon A. Rieger[5] and Adam Bourassa[5]

[1]Finnish Meteorological Institute, Helsinki, Finland
[2]Uviversity of Bremen, Bremen, Germany
[3] ESA-ESRIN, Frascati, Italy
[4] NASA Langley Research Center, Hampton, VA, USA
[5] Institute of Space and Atmospheric Studies, University of Saskatchewan, Saskatoon, Canada

*Correspondence to*: Viktoria F. Sofieva (viktoria.sofieva@fmi.fi)

**Abstract.** Climate-related studies need information about the distribution of stratospheric aerosols, which influence the energy balance of the Earth's atmosphere. In this work, we present a merged dataset of vertically resolved stratospheric aerosol extinction coefficients, which is derived from data by six limb and occultation satellite instruments: SAGE II on ERBS, GOMOS and SCIAMACHY on Envisat, OSIRIS on Odin, OMPS on Suomi-NPP, and SAGE III on the International Space Station.

The merging of aerosol profiles is performed by transformation of the aerosol datasets from individual satellite instruments to the same wavelength 750 nm and their de-biasing and homogenization by adjusting the seasonal cycles. After such homogenization, the data from individual satellite instruments are in good agreement. The merged aerosol extinction coefficient is computed as the median of the adjusted data from the individual instruments.

The merged time series of vertically resolved monthly mean aerosol extinction coefficients at 750 nm is provided in 10° latitudinal bins from 90°S to 90°N, in the altitude range from 8.5 km to 39.5 km. The time series of the stratospheric aerosol optical depth (SAOD) is created by integration of aerosol extinction profiles from the tropopause to 39.5 km; it is also provided as monthly mean data in 10° latitudinal bins. The created aerosol climate record covers the period from October 1984 until May 2022, and it is intended to be extended in the future. It can be used in various climate-related studies.

## 1    Introduction

Stratospheric aerosols play an important role in the Earth system and in the climate.  Through the scattering of solar radiation back to space and by heating the stratosphere through the absorption of thermal infrared radiation upwelling from the troposphere, stratospheric aerosols impact the radiative forcing and thus the energy balance of the Earth's atmosphere. By providing a surface for heterogeneous reactions, which release halogens, stratospheric aerosols contribute to the catalytic depletion of ozone. Because of a strong coupling between stratospheric aerosols, stratospheric ozone amount, and thermal





balance and dynamics of the atmosphere, it is essential to consider realistic aerosol information in modelling studies and interpretation of the measurements related to the stratosphere. The information about stratospheric aerosols and their influence on climate is also of value for investigations and analyses relevant for geoengineering (e.g., Crutzen, 2006).

One of the characteristics widely used to describe the amount of stratospheric aerosol is its extinction coefficient[1]. As
discussed e.g. by von Savigny et al. (2015), at a first approximation the aerosol extinction coefficient can be used to estimate the radiative forcing and thus quantify the impact of aerosols on climate change. There are several data sets of the stratospheric aerosol extinction coefficient retrieved from satellite measurements in limb-viewing geometry (e.g., Bourassa et al., 2012; Damadeo et al., 2013; von Savigny et al., 2015; Vanhellemont et al., 2010; Loughman et al., 2018). The first joint analysis of aerosol records from multiple instruments has been done by Vernier et al. (2011), but this study did not aim at creating a long-
term merged data set. A merged time series of the aerosol extinction coefficient using SAGE (Stratospheric Aerosol and Gas Experiment) II and OSIRIS (Optical Spectrograph and InfraRed Imaging System) aerosol data was presented by Rieger et al. (2015). The merging approach was based on a simple scaling of the extinctions at different wavelengths assuming a prescribed wavelength dependence characterized by a fixed value of the Ångström exponent. The latter represents a drawback of the method as the Ångström exponent might vary significantly even in the absence of very strong volcanic eruptions (e.g., Malinina
et al., 2019). The GloSSAC aerosol climatology presented by Thomason et al. (2018) is an extension of the climatology described by Rieger et al. (2015) including the earlier observations from 1979 to 1984, extending the coverage of the data set during the SAGE II measurement period and adding CALIPSO (Cloud-Aerosol Lidar and Infrared Pathfinder Satellite Observation) observations after 2006. This climatology uses, however, the same conversion of the OSIRIS data to SAGE II wavelengths as described by Rieger et al. (2015). The updated version of GloSSAC (Kovilakam et al., 2020) includes the latest
versions of the individual aerosol dataset and added SAGE III aerosol extinction on International Space Station (ISS), which allows an improved conversion of aerosol extinction to other wavelengths to be made.

This paper describes a merged long-term dataset of aerosol extinction coefficient profiles at 750 nm derived from data by solar/stellar occultation and limb-scatter instruments: SAGE II on ERBS (Earth Radiation Budget Satellite), GOMOS (Global Ozone Monitoring by Occultation of Stars) and SCIAMACHY (Scanning Imaging Spectrometer for Atmospheric
Chartography) on Envisat, OSIRIS on Odin, OMPS-LP (Ozone Monitor Profiling Suite Limb Profiler) on Suomi-NPP, and SAGE III on ISS (International Space Station). This dataset is referred to as Climate Data Record of Stratospheric Aerosols (CREST) hereafter. Our objective is to increase the reliability of the data set by including the extinction measurements from multiple instruments measuring similar atmospheric quantities in the limb-viewing geometry in the post-SAGE II period The paper is organized as follows. Section 2 is dedicated to short descriptions of aerosol datasets from individual satellite
instruments. Section 3 presents the merging algorithm. The correction and filling gaps on SAGE II data during the high

---

[1] The extinction coefficient is the sum of absorption and scattering in the atmosphere along the radiation path per unit distance. It is equal to $\sum N_i(\sigma_{a,i}(\lambda) + \sigma_{s,i}(\lambda))$, where $N_i$ is the number density of atmospheric constituent $i$, $\sigma_{a,i}(\lambda)$ is the absorption coefficient of species $i$ at wavelength $\lambda$, $\sigma_{s,i}(\lambda)$ is the scattering coefficient of species $i$ at wavelength $\lambda$.





volcanic load after the Pinatubo eruption is discussed in Section 4. Creating stratospheric optical depth is discussed in Section 5. Illustrations of the merged dataset and its comparison with GloSSAC v2.21 are presented in Section 6. Summary and discussion (Section 7) conclude the paper.

## 2    Satellite aerosol data

For creating the CREST merged dataset, we used aerosol profiles from several limb and occultation instruments. The information about individual datasets is summarised in Table 1. Below we present also the descriptions of individual aerosol datasets.

**Table 1. Information about the datasets used in the CREST dataset.**

| Instrument/ satellite | Processor, references | Time period | Local time | Vertical resolution | Profiles per day | Wavelength(s) for aerosol retrievals |
|---|---|---|---|---|---|---|
| SAGE II/ ERBS | NASA v7.0 (Damadeo et al., 2013) | Oct 1984 – Aug 2005 | sunrise, sunset | ~1 km | 14–30 | 386, 452, 525 and 1020 nm |
| OSIRIS/ Odin | USask v7.2 (Rieger et al., 2019) | Nov 2011 – present | 6 a.m., 6 p.m. | ~2 km | ~250 | 750 nm |
| GOMOS/ Envisat | FMI-GOMOSaero v.1 (Sofieva et al., 2023) | Aug 2002 – Dec 2011 | 10 p.m. | 3 km | monthly mean values | 400, 440, 452, 470, 500, 525, 550, 672 and 750 nm |
| SCIAMACHY/ Envisat | UBr v3.0 (von Savigny et al., 2015; Rieger et al., 2018; Malinina et al., 2019 | Aug 2002- Apr 2012 | 10 a.m. | ~3 km | ~1300 | 750 nm |
| OMPS-LP/ Suomi NPP | UBr v2.1 (Rozanov et al., 2024) | Apr 2012–present | 1:30 p.m. | ~2 km | ~1600 | 869 nm |
| SAGE III /ISS | NASA, AO3 v5.2 (Wang et al., 2020) | 2017 – present | sunrise, sunset | ~1 km | ~30 | 384, 449, 520, 602, 676, 756, 869, 1021, and 1544 nm |

## 2.1    OSIRIS

OSIRIS on board the Odin satellite(Llewellyn et al., 2004, p.200) performs limb-scatter measurements of the atmosphere from 2001 to present. In our work we use the cloud-cleared aerosol extinction profiles from new v7.2 retrievals (Rieger et al., 2019). The v7.2 processor uses wavelengths at 470, 675, 750 and 805 nm to retrieve aerosol extinction at 750 nm. The retrievals have reduced measurement geometry biases and improve extinction retrieval in the upper troposphere and lower stratosphere,

compared to the previous v5.10 retrieval algorithm.  The OSIRIS v7.2 aerosol extinction profiles are provided at 750 nm in the vertical range from ~6.5 km up to 35 km with the vertical resolution of ~ 2 km. According to the recommendations of the OSIRIS team, post-2022 data are not used in the merged dataset.

## 2.2    SAGE II

SAGE II was a solar occultation instrument that operated on board the ERBS satellite from 1984-2005

(https://sage.nasa.gov/missions/about-sage-ii/, Russell and McCormick, 1989). SAGE II had seven channels at ultraviolet, visible, and infrared wavelengths. The aerosol retrieval algorithm provides information about the size distribution, composition, and concentration of the aerosols (Thomason et al., 2008). In our analyses, we used v7 aerosol data. The detailed description of aerosol retrievals can be found in (Damadeo et al., 2013). SAGE II aerosol extinction profiles are provided at 386, 452, 525 and 1020 nm in the stratosphere and in the upper troposphere with the vertical resolution of ~ 1.0 km.

We applied the following cloud filtering of individual SAGE II profiles. The cloud altitude is defined at location where aerosol extinction at 1020 nm $\beta_{1020\,nm} \geq 2.5 \cdot 10^{-3}\ km^{-1}$  and ratio of extinctions $\frac{\beta_{525\,nm}}{\beta_{1020\,nm}} \leq 1.75$. Each profile was examined for presence of clouds; if found, all data at and below 1 km above that altitude are excluded.

## 2.3    SAGE III/ISS

 SAGE III is a solar occultation instrument currently operating on board the ISS; it was installed in 2017 (https://www-sage3-

iss.larc.nasa.gov/).  The SAGE III detectors cover from the ultraviolet to the near-infrared parts of the spectrum, and they allow for the characterization of different types of aerosols in the atmosphere. The retrieval methodology is similar to that of SAGE II, albeit with more spectral channels to better constrain the shape of the aerosol extinction spectrum (Wang et al., 2020). In our analyses, we used v5.3 aerosol data.

SAGE III/ISS aerosol extinction profiles are provided at 9 wavelengths (384, 449, 520, 602, 676, 756, 869, 1021, and 1544

nm) in the altitude range from cloud top to ~45 km with the vertical resolution of ~ 0.75 km. The SAGE III/ISS data are filtered for presence of clouds in the same way as SAGE II data.

## 2.4    SCIAMACHY

The SCIAMACHY instrument was part of the payload of the ESA Envisat (Burrows et al., 1995; Bovensmann et al., 1999) launched on the 28th of February 2002 into a sun-synchronous orbit, with an equator crossing time of 10:00 am in a descending

node. It made measurements, including those during its commissioning phase from March 2002 to April 2012 when Envisat failed. During each orbit, measurements in the following three observation modes were made: nadir, limb, and solar/lunar occultation. Of relevance to this study are the measurements made in the limb viewing geometry, which achieve global coverage in 6 days at the equator.  These are used to retrieve aerosol extinction profiles (von Savigny et al., 2015; Rieger et



al., 2018; Malinina et al., 2019). We note that, in addition, solar occultation measurements were also made (see Noel et al.,
2020) close to the terminator in the northern hemisphere but have limited latitude coverage.

In the limb measurement mode, the SCIAMACHY entrance optics, which include two scan mirrors and a telescope, collect
the solar radiance upwelling from the atmosphere. The latter comprises the radiance scattered towards SCIAMACHY, i.e., a)
that passes through the atmosphere en route to the surface and b) that is scattered back by the surface or clouds and then
scattered by the atmosphere. In the standard limb scan mode, as opposed to an alternative optimized for measurements in the
mesosphere and lower stratosphere, the SCIAMACHY optical scanner unit register vertical profiles of the scattered radiance
from a tangent point, close to or just below the surface, to a tangent height of ~100 km.

The vertical scanning was done with a sampling of about 3.3 km. The instantaneous field of view of the instrument was about
2.6 km. At each tangent height, a horizontal scan with typically 4 measurements and a total swath of 960 km was performed.
The spectral coverage of SCIAMACHY was 214 - 2386 nm with spectral resolution varying from 0.2 to 1.5 nm. In this study,
Level 1 data (Version 8) provided by ESA are used, along with the utilization of calibration flags 1, 2, 4, 5, and 7. These flags
address, respectively, the amount of leakage current, the pixel-to-pixel gain, stray-light correction, wavelength calibration, and
radiometric calibration.

The aerosol extinction coefficients used in this study are from the University of Bremen processor V3.0, which is a further
development of the V1.4 processor described by Rieger et al. (2018). Similar to the V1.4 aerosol extinction product, the
retrieval is performed at 750 nm. The retrieval is done iteratively using the Levenberg-Marquardt approach. The regularization
is applied to the relative differences with respect to the prior profile with a priori standard deviation set to 50. The main
differences between V3.0 and V1.4 include the use of sun-normalized radiance instead of normalization to a limb measurement
at an upper tangent height and the utilization of an effective surface albedo retrieved from collocated nadir measurements. The
latter approach is similar to that used by Pohl et al. (2023), however, the effective surface albedo is based solely on the
collocated nadir observations and is not changed any more when retrieving the aerosol extinction profile. In addition, the
vertical range of the retrieval is extended to 9 – 38 km in comparison with 12 - 35 km in V1.4. The validation results for V3.0
extinction retrieval show that the agreement with SAGE II data is similar to that found with V1.4 data products. However,
most of the outliers identified in the comparison of V1.4 extinction data with independent measurements are not seen in the
comparisons for the V3.0 product anymore.

## 2.5    GOMOS

GOMOS is a stellar occultation instrument operated on board Envisat satellite in 2002-2012 (Bertaux et al., 2010; Kyrölä et
al., 2004). GOMOS measured stellar light in the occultation geometry from above the atmosphere until the star lost in the field
of view, typically at altitudes 10-15 km. In our analyses, we use the aerosol data retrieved with the FMI-GOMOSaero v1.0
processor (Sofieva et al., 2023), which uses monthly and zonally averaged nighttime transmittance spectra as a basis for aerosol
retrievals. The inversion algorithm uses the same strategy as GOMOS retrievals of trace gases: the spectral inversion followed
by the vertical inversion (Kyrölä et al., 2010). The spectral inversion relies on removal of contribution from ozone, $NO_2$, $NO_3$





and Rayleigh scattering from the optical depth spectra, for each ray perigee altitude. The FMI-GOMOSaero dataset v.1.0 provides aerosol extinction profiles in the altitude range 10-40 km at wavelengths 400, 440, 452, 470, 500, 525, 550, 672 and 750 nm with a vertical resolution of 3 km.

## 2.6 OMPS-LP

OMPS-LP is a part of the OMPS instrument suite operating on board the Suomi-NPP satellite of NOAA/NASA since the end of 2011 (Flynn et al., 2014). The field of view of the OMPS-LP instrument is pointed tangentially to the Earth's surface (limb-viewing geometry) and collects spectral radiance scattered by the Earth's atmosphere and reflected by the underlying surface by using a two-dimensional charged-coupled device (CCD). The atmosphere is observed in the tangent height range from 0 to 100 km with a vertical sampling of 1 km and vertical field of view of each pixel of about 1.5 km. The spectral coverage of the OMPS-LP instrument is 280 - 1000 nm with a spectral resolution increasing from 1 nm in the UV region to about 30 nm in the near-IR. In this study, Level 1 data V2.6 provided by NASA are used. Only measurements from the central slit are considered.

The aerosol extinction coefficients used in this study are from the University of Bremen product V2.1, which is described in details in Rozanov et al. (2024). Briefly, the aerosol extinction coefficients are retrieved at a wavelength of 869 nm using the sun-normalized radiance in the tangent height range from 8.5 to 48.5 km. The retrieval is iterative and uses the Levenberg-Marquardt approach with Tikhonov regularization (zeroth order term and first order derivative). At each iteration, the vertical profile of the aerosol extinction coefficient and the effective surface albedo are estimated simultaneously using all measurements in the selected tangent height range. The regularization is done with respect to the solution at the previous iteration.

## 3 Merged dataset of aerosol profiles

The datasets from individual satellite instruments – GOMOS, OMPS-LP, OSIRIS, SAGE II, SAGE III/ISS, SCIAMACHY– were collected and gridded as monthly zonal means with $10^{\circ}$ step in the latitude. OSIRIS dataset was selected as the reference because it has the longest data record. As OSIRIS dataset is available only at 750 nm, this wavelength was chosen as the reference wavelength. The extinction data, retrieved from the observations of the other instruments, were converted to the wavelength of 750 nm, if not already retrieved at this wavelength. For the conversion of SAGE II data, we used aerosol extinction at 525 nm and 1020 nm. The conversion is based on the Ångström formula

$$\beta_\lambda = \beta_{\lambda_0} \left(\frac{\lambda}{\lambda_0}\right)^{-\alpha}, \tag{1}$$

where $\beta_\lambda$ and $\beta_{\lambda_0}$ are aerosol extinction coefficients at wavelengths $\lambda$ and $\lambda_0$, and $\alpha$ is the Ångström exponent, with a correction derived from SAGE III/ISS data (for details, see Damadeo et al., 2023):

$$\beta_{750\,nm} = \beta_{1020\,nm} \left(\frac{750}{1020}\right)^{-\alpha} (1.23 - 0.055\alpha). \tag{2}$$



In Eq.(2), the Ångström exponent α is determined using the SAGE II aerosol extinction data at 525 and 1020 nm. . This correction accounts for the bias introduced when using the Angstrom exponent for interpolation as the aerosol extinction spectrum does not quite follow the Angstrom formula.

GOMOS 675-nm aerosol data were converted to 750 nm using the Ångström exponent, which was determined from GOMOS 500 and 675 nm aerosol extinction coefficients. As no measurement data providing the Ångström exponent for the entire operation period of OMPS-LP are available, the Ångström exponent used to convert OMPS-LP data was obtained by Mie calculations assuming a fixed particle size distribution, PSD ($r_{med}$ = 0.08 μm, σ = 1.6). When scaling from 869 to 750 nm wavelength this corresponds to a constant factor of 1.477. The fixed PSD assumed here is the same as used in the OMPS-LP

retrieval procedure and based on an arbitrarily selected balloon-borne in-situ measurement presented by Deshler (2008). Gridded time series from individual instruments were compared and methods to remove biases between the time series were analyzed.

       SCIAMACHY, OMPS-LP and GOMOS are adjusted as follows. For each instrument *i*, latitude zone and altitude level, the seasonal cycle $S_i(m)$, *m* is a month, is estimated using volcanic-free periods (years 2001-2005, 2007, 2010, 2013-

2016, 2020). Then the seasonal cycles are fitted to the OSIRIS seasonal cycle using a robust linear regression

$$S_{OSIRIS}(m) = a + b \cdot S_i(m) + \epsilon \tag{3}$$

In Eq.(3), *a* and *b* are the fitted coefficients and $\epsilon$ is noise. This procedure ensures that the scaling is performed properly even in cases where not full year is covered by OISRIS data. The original seasonal cycles of SCIAMACHY, OMPS-LP and GOMOS are replaced with fitted seasonal cycles:

$$\beta_{corr,i} = \beta_i - S_i + (a + b \cdot S_i), \tag{4}$$

where $S_i$ is the values for the seasonal cycle at the corresponding month.

       For SAGE II and SAGE III/ISS, due to limited data sampling and a relatively short overlapping period with OSIRIS, the seasonal cycle correction is not performed. For these datasets, the offset is computed as the median difference between OSIRIS and occultation aerosol data in their overlapping periods (the period is 2017-2021 for SAGE III and 2001-2005 for

SAGE II) and added to data. The adjustment procedure works well, also in complicated cases.

       The merged aerosol extinction coefficient time series is calculated as the median of the adjusted data from the individual instruments. Taking the median has the advantage of minimising the influence of outliers that might be present in the individual datasets.

Illustrations of original, adjusted and merged datasets in several latitude zones are presented in Figures 1 and 2, as well as in

Figures S1-S4 of the Supplement. These figures show that the adjustment procedure makes the datasets from individual datasets much closer to one another in period of overlapping measurements.

The uncertainty of the merged aerosol profiles are estimated similar to the approach used by Sofieva et al. (2017). For individual datasets, the uncertainty of the monthly mean values is estimated as the standard error of the mean, i.e., $\sigma_\beta^2 = \frac{s^2}{N}$, where $s^2$ is the sample variance and *N* is the number of measurements. The uncertainties of data adjustments are approximately



evaluated as uncertainty of bias computation and added quadratically to all individual monthly mean records except that of OSIRIS. The uncertainty of the merged aerosol extinction (which correspond to the median value $\beta_{merged}$) is estimates as (Sofieva et al., 2017):

$$\sigma_{\beta,merged} = min\left(\sigma_{\beta,i_{med}}, \sqrt{\frac{1}{N}\sum_{i=1}^{N}\sigma_{\beta,i}^2 + \frac{1}{N^2}\sum_{i=1}^{N}\left(\beta_{corr,i} - \beta_{merged}\right)^2}\right), \quad (5)$$

where $\sigma_{\beta,i_{med}}$ is the uncertainty of the extinction corresponding to the median value.

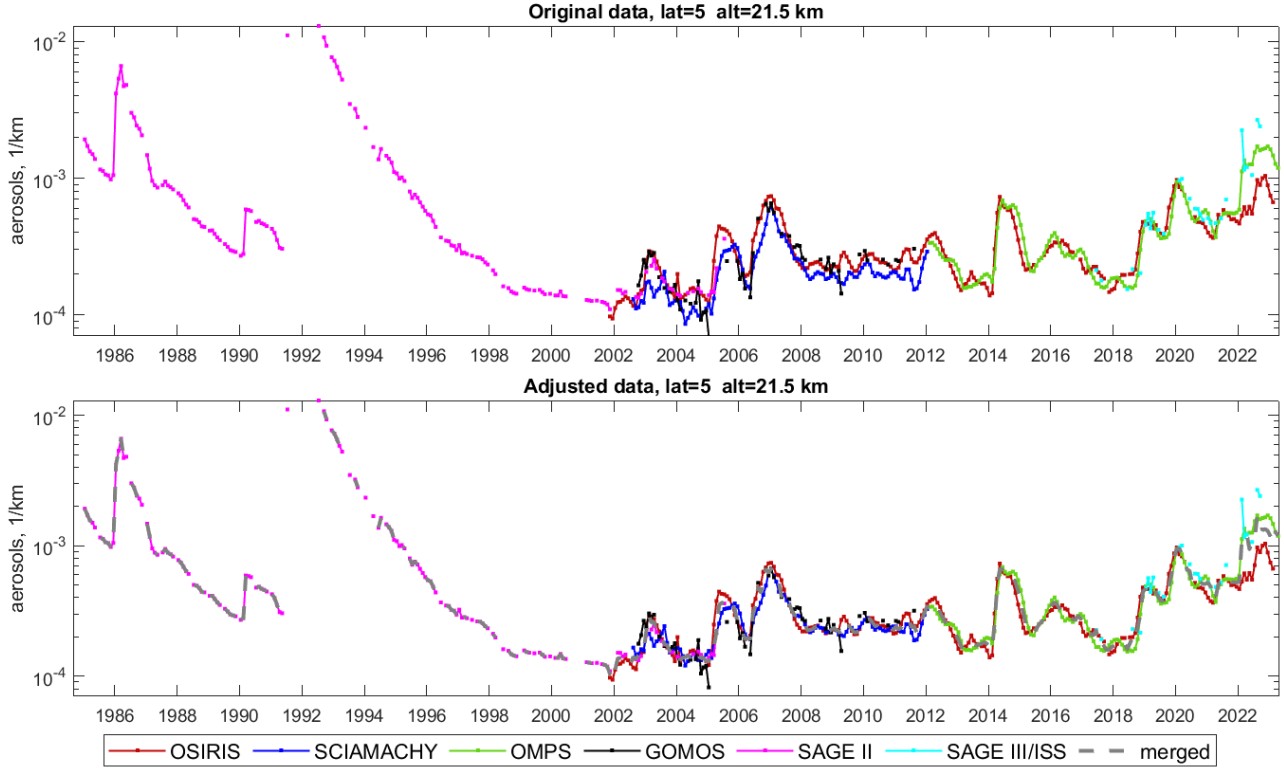


**Figure 1. Aerosol extinction coefficient (1/km) at 21.5 km in the latitude zone 0–10°N. Colored lines correspond to aerosol records from individual instruments. Top panel: original data, bottom panel: adjusted data. The merged aerosol extinction coefficient is shown by grey line in the bottom panel.**

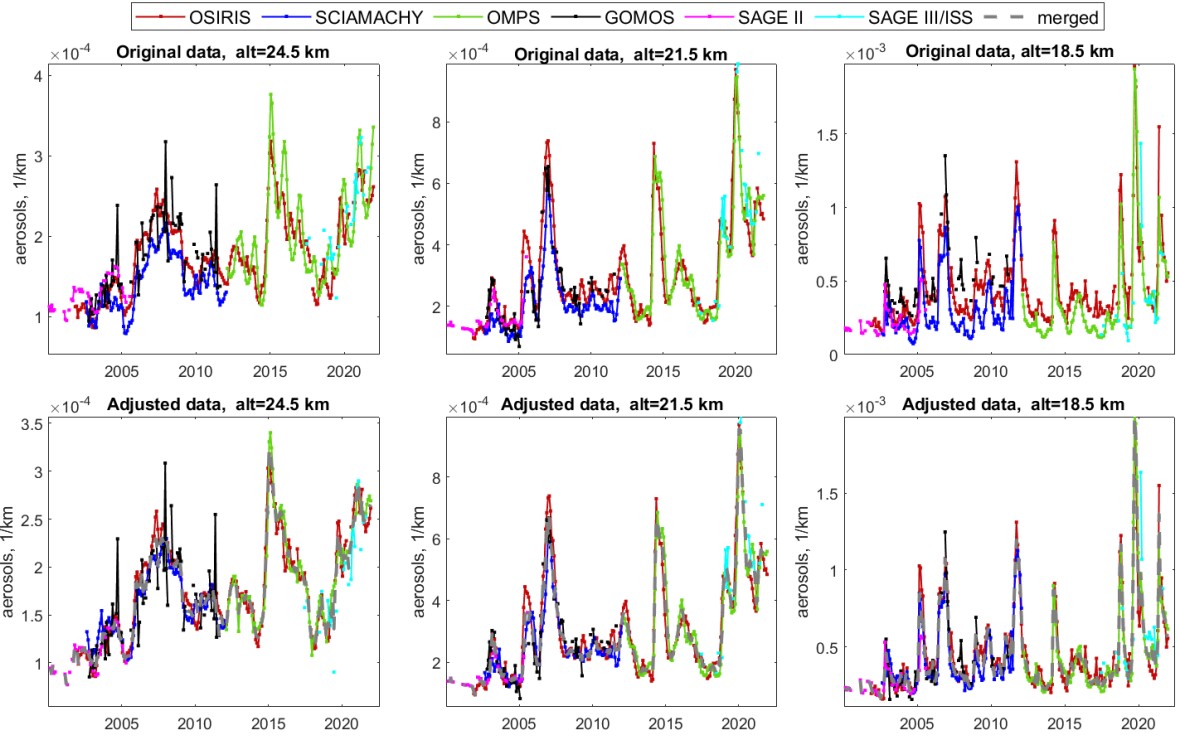

**Figure 2. As Figure 1, but for several altitude levels and with the zoom on years 2000–2021.**

Figure 3 shows the aerosol extinction profiles from the merged CREST dataset in three latitude zones, 30°–60°S (top), 20°S–20°N (center) and 30°–60°N (bottom). The main volcanic eruptions and strong wildfires are indicated in the figure with a bar of length proportional to the volcanic explosivity index (VEI). As observed in Figure 3, enhancements of aerosol extinction correspond to the enhanced volcanic activity or strong wildfires, which are listed in Table 2.



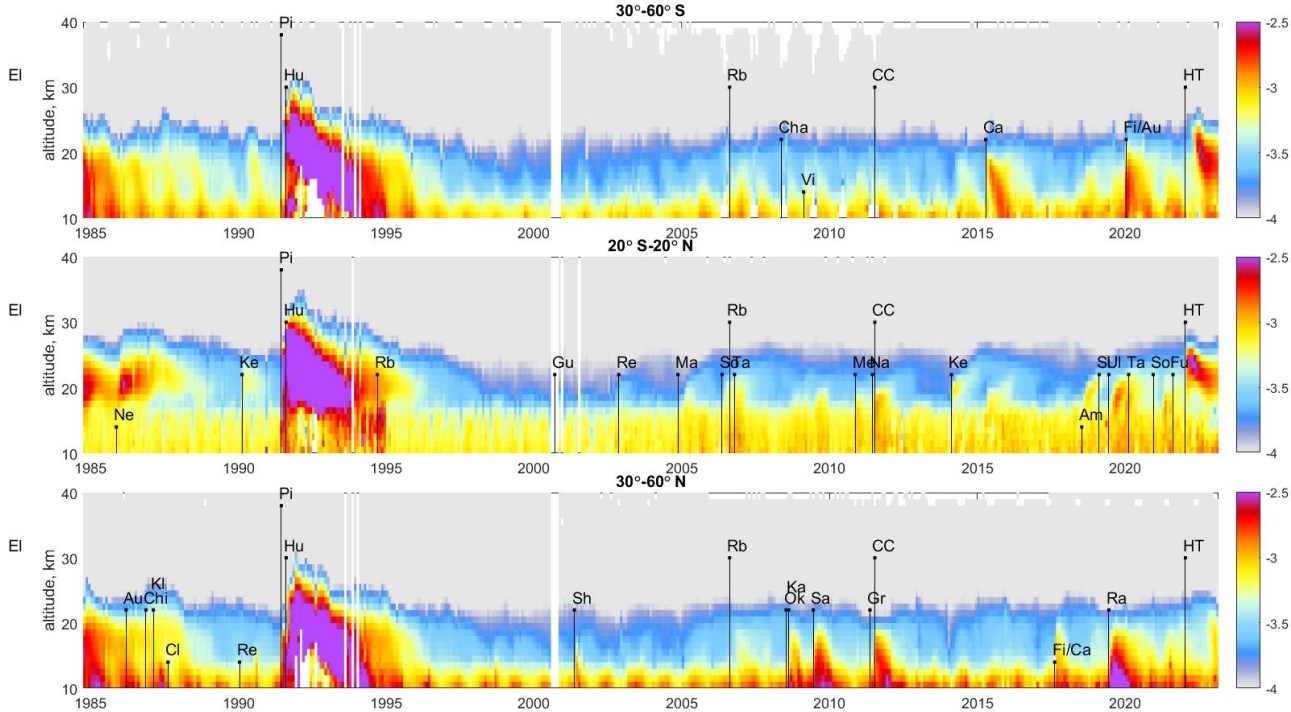

Figure 3. The CREST merged aerosol extinction profiles at 30°–60°S (top), 20°S–20°N (center) and 30°–60°N (bottom). The volcanic eruptions are indicated by black bars with the length being proportional to the values of volcanic explosivity index (VEI). Volcanos with VEI>=5 are shown for all latitude zones, and with VEI>3 in the corresponding latitude zones.

Table 2. The list of volcanic eruptions and strong wildfires, for which rise to the stratosphere.

| Year | Month | Volcano/wildfire name | Abbreviation | Latitude (deg North) | VEI |
|------|-------|----------------------|--------------|----------------------|-----|
| 1985 | 11 | Nevado del Ruiz | Ne | 5 | 3 |
| 1986 | 3 | Augustine | Au | 59 | 4 |
| 1986 | 11 | Chikurachki | Chi | 50 | 4 |
| 1987 | 2 | Kliuchevskoi | Kl | 56 | 4 |
| 1987 | 8 | Cleveland | Cl | 53 | 3 |
| 1990 | 1 | Redoubt | Re | 61 | 3 |
| 1990 | 2 | Kelut | Ke | -8 | 4 |
| 1991 | 6 | Pinatubo | Pi | 15 | 6 |

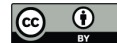



| 1991 | 8 | Mt.Hudson | Hu | -46 | 5 |
|---|---|---|---|---|---|
| 1994 | 9 | Rabaul | Rb | -4 | 4 |
| 1999 | 8 | Sheveluch | Sh | 57 | 4 |
| 2000 | 9 | Guagua Pichincha | Gu | 0 | 4 |
| 2001 | 5 | Shiveluch | Sh | 56 | 4 |
| 2002 | 11 | Reventador | Re | 0 | 4 |
| 2004 | 11 | Manam | Ma | 4 | 4 |
| 2006 | 5 | Soufrière Hills | So | 16 | 4 |
| 2006 | 10 | Rabaul/Tavurvur | Ta | -4 | 4 |
| 2008 | 5 | Chaitén | Cha | -42 | 4 |
| 2008 | 7 | Okmok | Ok | 55 | 4 |
| 2008 | 8 | Kasatochi | Ka | 55 | 4 |
| 2009 | 2 | Fire/Victoria | Vi | -37 | 3 |
| 2009 | 6 | Sarychev | Sa | 48 | 4 |
| 2010 | 11 | Merapi | Me | -8 | 4 |
| 2011 | 5 | Grimsvótn | Gr | 64 | 4 |
| 2011 | 6 | Nabro | Na | 13 | 4 |
| 2011 | 7 | Cordon Caulle | CC | -40 | 5 |
| 2014 | 2 | Kelut | Ke | -8 | 4 |
| 2015 | 4 | Calbuco | Ca | -41 | 4 |
| 2017 | 8 | Wildfires/California | Fi/Ca | 51 | 3 |
| 2018 | 7 | Ambae | Am | -15 | 3 |
| 2019 | 2 | Sinabung | Si | 3 | 4 |
| 2019 | 6 | Ulawun | Ul | -5 | 4 |
| 2019 | 6 | Raikoke | Ra | 48 | 4 |
| 2020 | 2 | Taal | Ta | 14 | 4 |
| 2020 | 1 | Fire/Australia | Fi/Au | -35 | 4 |
| 2020 | 12 | Soufriere St.Vincent | So | 13 | 4 |
| 2021 | 8 | Fukutoku-Oka-no-Ba | Fu | 24 | 4 |
| 2022 | 1 | HungaTonga-Hunga | HT | -21 | 5 |




## 4   Filling gaps in SAGE II data during Pinatubo

It has been reported in several publications that SAGE II aerosol load is underestimated or missing in the lower stratosphere during Pinatubo volcanic eruption (Kovilakam et al., 2020; Sukhodolov et al., 2018). The reason is "saturation" in SAGE II data, i.e., there are no values larger than ~0.012 km$^{-1}$. The GloSSAC dataset (Kovilakam et al., 2020; Thomason et al., 2018)

uses experimentally scaled CLAES data at these times/locations. Instead, we applied the following approach in the merged CREST dataset. In the period affected by Pinatubo volcanic eruption, i.e., from June 1991 and lasting 3 years, the aerosol extinction $\beta(t)$ as a function of time for each altitude level is fitted by the function

$$f(t) = a + b \left(1 - \exp\left(-\frac{t}{\tau_1}\right)\right) \exp\left(-\frac{t}{\tau_2}\right), \tag{5}$$

where $a$, $b$, $\tau_1$ and $\tau_2$ are parameters to fit. Eq.(5) gives a simple parameterization of fast growing and gradual decrease; it is

not linked with a dynamical model. The coefficient $a$ is estimated as the mean over the period from January to May 1991, while other parameters are obtained via non-linear least-squares fitting by the Levenberg-Marquardt algorithm. As illustrated in Figure 4 (left) for 27.5 km, the aerosol peak associated with the Pinatubo eruption follows Eq. (5) well, for the cases where SAGE II data are not saturated. For lower altitudes (from the tropopause to the level 8 km above the tropopause), the data from 3 to 12 months after the Pinatubo eruption are not used in the fit, as illustrated in Figure 4 (right). The fitted aerosol peak

during the strong Pinatubo load exceeds the maximal aerosol extinction value of ~0.012 km$^{-1}$ reported by SAGE II. The period for omitting data, from 3 to 12 months after Pinatubo eruption, was selected after visual inspection, it corresponds to typical saturation period of nearly constant values. The fitting is performed in the stratosphere only, where there are a sufficient number of data points.

We performed such Pinatubo-filtering to the data in the tropics (20°S − 20°N) and in other latitude zones (20°− 40°, 40°− 60°,

60°− 90°, both North and South), and found that the saturation primarily affects the data in the tropics. Therefore, for latitude zones from 20°S to 20°N the saturated aerosol data (i.e., for altitudes below 26 km, where the fitted aerosol value exceeds 0.011 km$^{-1}$) are replaced with the fitted values. Missing data during Pinatubo eruption are also replaced with the fit.



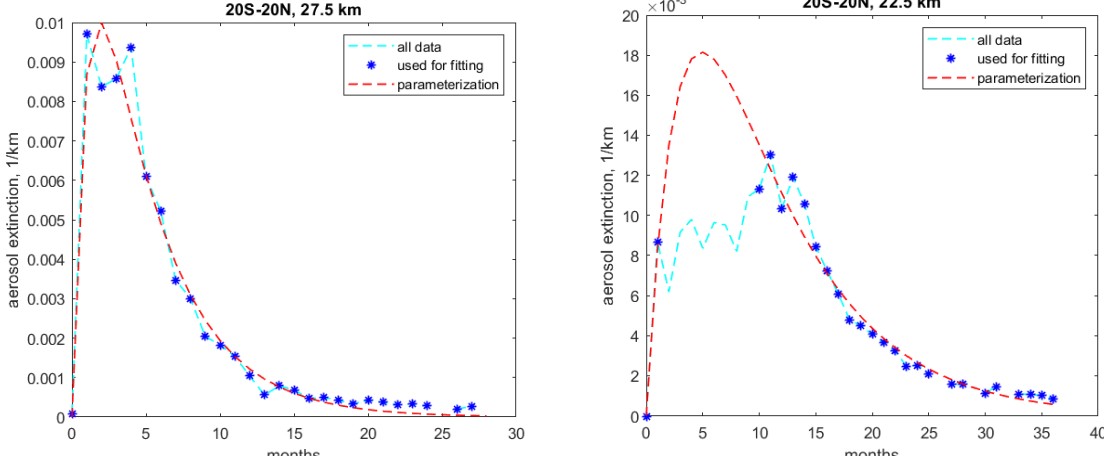

**Figure 4. Example of the SAGE II aerosol fitting during the Pinatubo period with the function Eq.(5) at 27.5 km (left) and 22.5 km (right). The data are for latitudes 20°S–20°N. For 27.5 km, all data used for the fit. For 22.5 km, the data from 3 to 10 months after Pinatubo eruption are excluded.**

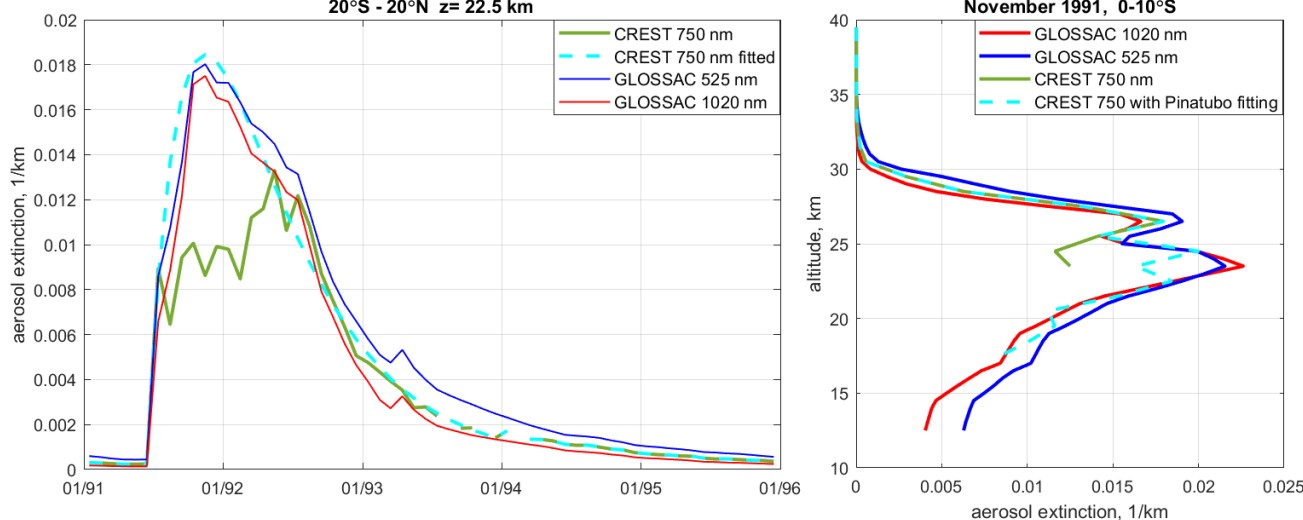

**Figure 5. Left: time series of aerosol extinction at 22.5 km as reported by GloSSAC (at 525 and 1020 nm) and CREST (original data (green solid line) and fitted by Eq.(5), cyan dashed line). Right: GloSSAC, CREST original and Pinatubo-corrected aerosol profiles in November 1991, in the latitude zone 0°–10°S.**

The resulting Pinatubo-corrected aerosol profiles agree with GloSSAC within 5-10%, as illustrated in Figure 5.



## 5 Creating stratospheric aerosol optical depth


The stratospheric aerosol optical depth (SAOD) has been computed via integration of aerosol extinction profiles from tropopause to ~40 km. The tropopause is computed using ERA-5 data according to the WMO definition of thermal tropopause. The resulting SAOD record is shown in Figure 6(a).

**Figure 6.Illustration of creating CREST SAOD and aerosol proxy. (a) SAOD as obtained from the integration of aerosol profiles; (b) as (a) but missing data are filled in with the 3-month mean, for each latitude zone; (c) the gap-free SAOD obtained from (b) by interpolation by Delaney triangulation; (d) global mean SAOD (aerosol proxy). All SAOD values are plotted using logarithmic scales.**



The uncertainty of the stratospheric aerosol optical depth is evaluated using the error propagation.

For using SAOD in climate simulations and in trend analyses, a gap-free dataset is needed. For this, we first fill small gaps by smoothing the data in each latitude zone by applying a 3-month running average (Figure 6b). Then the Delaunay triangulation method is applied to the time-latitude SAOD field, resulting in gap-free stratospheric aerosol climate data record, which is illustrated in Figure 6c.

The global average (area-weighted) in the latitude range from 80°S − 80°N stratospheric aerosol optical depth is shown in Figure 6(d); it is used as aerosol proxy in e.g. trend analyses.

## 6    The merged CREST SAOD dataset: illustrations, comparison with GLOSSAC

The interpolated CREST SAOD dataset is shown in Figure 7(a), together with main aerosol events (volcanic eruptions and wildfires), which are indicated by black circles of the size proportional to their volcanic explosivity index VEI (Table 1). As

observed in Figure 7(a), aerosol enhancements match very well with volcanic eruptions and strong wildfires.

The comparison of CREST and GloSSAC stratospheric optical depth datasets is shown in Figure 7. GloSSAC provides SAOD at 525 nm and 1020 nm, therefore SAOD is higher for 525 nm and lower for 1020 nm than CREST SAOD. The overall morphology of SAOD enhancements is very similar (taking into account the wavelength-dependence) in CREST and

GloSSAC. The comparison of global mean SAOD from CREST (750 nm) and GloSSAC (525 nm and 1020 nm) is shown in Figure 8. The global mean was computed as area-weighted SAOD in latitude range from 80° S to 80°N. In this figure, we show also global GloSSAC SAOD converted to 750 nm by using the same method as applied to SAGE II data (see Sect. 3).

As seen in Figure 8, CREST and GloSSAC aerosol proxies are in very good agreement and show expected wavelength dependence of aerosol extinction. At the peak of Pinatubo the CREST global SAOD is slightly different from that converted

from the GloSSAC SAOD, which is probably related to different fitting/scaling of SAGE II data used in CREST and GloSSAC.



**Figure 7. CREST SAOD at 750 nm (top), GloSSAC SAOD at 525nm (center), and GloSSAC SAOD at 1020 nm (bottom). SAOD is presented in logarithmic color scale. Volcanic eruptions and wildfires are indicated by black circles of the size proportional to their volcanic explosivity index VEI (Table 1).**



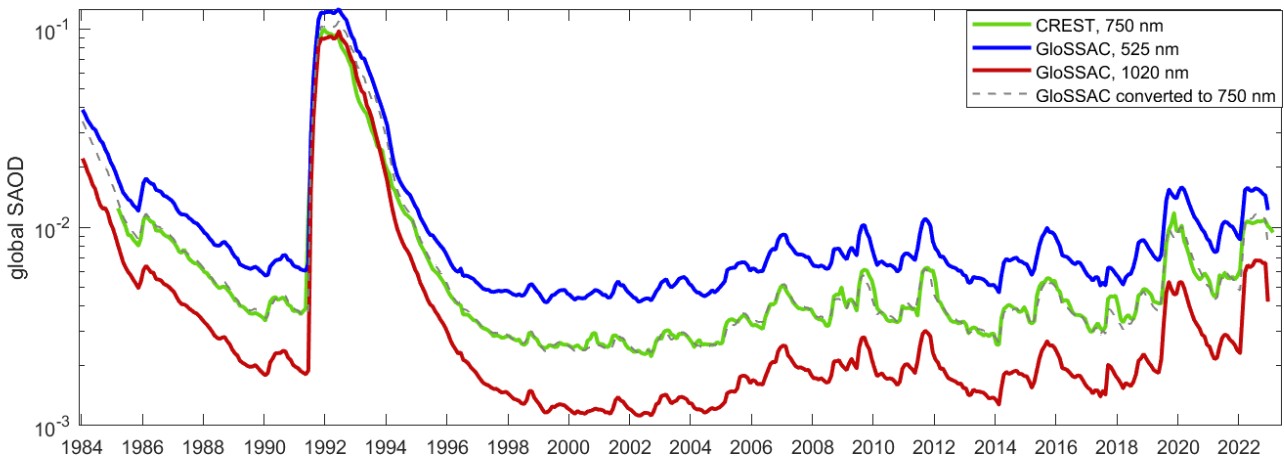

**Figure 8. Global mean stratospheric optical depth from CREST (750 nm) and GloSSAC (525 and 1020 nm).**

## 7    Summary and discussion

In this paper, we presented a new climate data record of stratospheric aerosols (CREST). The merged CREST dataset includes:

- Monthly zonal mean aerosol extinction profiles at 750 nm from 8.5 to 39.5 km in 10° latitudinal bins from 90°S to 90°N
- Monthly zonal mean stratospheric aerosol optical depth at 750 nm in 10° latitudinal bins from 90°S to 90°N
- Global mean stratospheric optical depth, which can be used as an aerosol proxy in trend analyses.

The CREST data set combines the data from six limb and occultation instruments: OSIRIS, SAGE II, SAGE III/ISS, SCIAMACHY, OMPS-LP, and GOMOS. The CREST dataset is complementary to NASA-GloSSAC: it provides aerosols at a different wavelength, 750 nm, it uses a different collection of individual datasets included in the merged dataset and a different merging approach. In addition, a different correction of saturated SAGE II data during Pinatubo period is applied in construction of the CREST dataset.  The overall agreement between the CREST and GloSSAC SAOD is very good.

The CREST dataset is in open access at https://fmi.b2share.csc.fi/records/8bfa485de30840eba42d1d407f4ce19c.  At the moment, it covers the period from 1984 to 2022, and it is intended to be extended in the future.

The created merged dataset of aerosol profiles can be used in various climate-related studies. For example, it can be used as a proxy in the regression models for trend analyses, or as a forcing in simulations with chemistry-transport models.

For the CREST dataset, we used aerosol extinction profiles from instruments measuring in the limb-viewing
geometry, and compared it with GloSSAC, which uses similar measurements. When new limb aerosol extinction profiles will be available, they can be easily added to the merged dataset using the methods explained in the paper.  However, there are also several        active        remote        sensing        instruments,        such        as        CALIPSO        and        Aeolus



(https://www.esa.int/Applications/Observing_the_Earth/FutureEO/Aeolus, Flament et al., 2021), and the planned mission EarthCARE (https://www.esa.int/Applications/Observing_the_Earth/FutureEO/EarthCARE ), which provide aerosol profiles from measurements in the nadir-viewing geometry. The quantities measured by active instruments (backscattering ratio) and sampling (about a hundred meters wide footprint stripe vs. several hundred kilometers large footprints of limb instruments) are very different to those from the instruments used in this study. These differences in the measurement principles result in different sensitivity to aerosols and vertical range of retrieved aerosol profiles. Nevertheless, comparison of aerosol climate data records from limb and nadir measurement systems might be an interesting and challenging future work.

**Data availability**

The merged CREST aerosol dataset (Sofieva, et al., 2022) is available at https://fmi.b2share.csc.fi/records/8bfa485de30840eba42d1d407f4ce19c. Interested readers can ask for the computational code from the corresponding author.

**Competing interests**

The contact author has declared that none of the authors has any competing interests.

**Acknowledgements**

The work is performed in the framework of ESA project CREST. The creation of the SCIAMACHY and OMPS-LP aerosol data sets at the University of Bremen was funded in parts by ESA via CREST project, by the German Research Foundation (DFG) via the Research Unit VolImpact (grant no. FOR2820), and by the University and State of Bremen. The University of Bremen team gratefully acknowledges the computing time granted by the Resource Allocation Board and provided on the supercomputer Lise and Emmy at NHR@ZIB and NHR@Göttingen as part of the NHR infrastructure. The calculations for this research were conducted with computing resources under the project hbk00098. The FMI team thanks the Academy of Finland (Centre of Excellence of Inverse Modelling and Imaging; decision 353082).

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
