# Peer review of "A Climate Data Record of Stratospheric Aerosols"

_Earth System Science Data, 2023_

## Author Response (AR1)

Dear Editor,

Thank you very much for your attention to our paper. We thank the reviewers for their valuable comments on our paper. Their comments are taken into account in the revised manuscript.

Please find below the response letters to the reviewers' comments (the same letters are posted in the interactive discussion). The revised manuscript with the modifications marked by "tracked changes" is also uploaded.

In addition, the CREST dataset is now extended until December 2023.

As a corresponding author, I confirm that all co-authors concur with the submission in its revised form.

Yours sincerely,

Viktoria Sofieva, Dr., Adj. Prof.
Finnish Meteorological Institute, Earth Observation
P.O. Box 503 (Erik Palmenin aukio, 1)
FIN-00101 Helsinki Finland
tel: +358 29 539 4698
fax: +358 29 539 3146
email: viktoria.sofieva@fmi.fi

**Review#1**

Dear Reviewer,
Thank you very much for your positive evaluation of our paper. We took your comments into account in the revised version of the manuscript. Please find below our detailed replies (black font) on your comments (blue font).
Reviewer#1 comments:
….In my opinion, the use of the merged CREST dataset is innovative and of high interest to the field. It is a useful tool which can be used in various climate-related studies. I agree with the authors that CREST can be used as a proxy in the regression models for trend analyses, or as a forcing in simulations with chemistry-transport models. At the moment, CREST covers the period from 1984 to 2022, and it is intended to be extended in the future. The use of new instruments could provide added value to the existed dataset. Overall, the manuscript is clear and well-structured giving a detailed analysis of the methodology that has been followed. Below I give some specifying remarks which can improve the manuscript.

**Specific comments:**

- **L85-87:** The implementation of the cloud-filtering to the SAGE II aerosol extinction profiles is not well supported. According to the authors, the cloud-affected altitudes are defined at locations where aerosol extinction values at 1020 nm are lower than $5 \cdot 10^{-3}\ km^{-1}$ and ratio of extinctions ($\beta_{525\ nm}/\beta_{1020\ nm}$) is lower than 1.75 without giving more detailed information for the selection of these threshold values.

There have been numerous different methods to attempt to filter SAGE (II, III/M3M, and III/ISS) data using retrieved aerosol extinction data over the past few decades. However, many of them tend to be variations on the same basic premise, namely looking at the ratio of extinctions between the 520 nm and 1020 nm channels to determine the likely presence of clouds (the closer the 520/1020 ratio is to 1, the more likely the result is a cloud) while simultaneously imposing some minimum extinction value to avoid false positives in the noise regime in the upper atmosphere. However, because observations tend to be a mixture of cloud and aerosol, there is no single definitive pairing of threshold values. Instead, different researchers have used different values leaning toward more or less conservative based on their purpose. For this study, using a ratio threshold of less than 1.75 is fairly conservative (as aerosol data within large volcanic eruptions and major pyrocbs can approach these values), while using an extinction threshold of greater than 2.5x10⁻³ km⁻¹ is less conservative to avoid accidentally omitting data from the Hunga Tonga eruption in the SAGE III/ISS data for this particular ratio threshold. This pair of threshold values is well within the range of other studies using similar simplistic cloud filtering algorithms.

- **L157-204:** It is quite difficult to read and understand exactly the implemented methodology for the aerosol dataset merging described in Section 3. There are a lot of phrases and equations involved which cause confusion to some degree. The authors are requested to seek a way to provide a clear description of the Section " Merged dataset of aerosol profiles" which is the key part of the analysis. Moreover, use two subsections instead of one whole section to describe the methodology giving more details in each section. The first one will be focused on the description of the conversion of the extinction coefficients from each instrument's frequency to the 750nm and the second one will describe the merging of aerosol datasets.

As suggested, we divided Sect. 3 into subsections and added some more details.

- In **Figures 1** and **2**, use the label "Extinction 1/km" instead of "aerosols 1/km". Add units in the plotted aerosol extinction profiles in Figure 3.

Corrected as suggested.

- **L259**: The authors are requested to give more information regarding the comparison of aerosol extinction time series and profiles between the CREST and GloSSAC datasets since this is the most significant part proving the reliability of the merged dataset.

In the revised version, we added: "Taking into account the uncertainty of gaps filling approaches during Pinatubo in GloSSAC and CREST, such an agreement can be considered as very good. Unfortunately, there are no available in-situ measurements of aerosol extinction profiles during this period to validate CREST and GloSSAC profiles. However, the similarity of GloSSAC and CREST aerosol profiles during the Pinatubo period increases confidence of both gap filling approaches."

- As already mentioned in **lines 316-324**, the comparison of CREST dataset using limb-viewing instruments with aerosol extinction profiles retrieved form active remote sensing instruments such as Aeolus, CALIPSO and the forthcoming EarthCARE satellite mission would provide added value increasing the reliability of the stratospheric aerosol climate data record. This is an interesting and challenging task which can be implemented in a future work (can be mentioned in the discussion or conclusion section).

We fully agree. Your suggested statement is added to Sect. 7 (Summary and discussion). We removed also "the planned mission" before EarthCARE, since the instrument is launched already.

**Review#2**

Dear Dr. Wilson,
Thank you very much for your positive evaluation of our paper.  We took your comments into account in the revised version of the manuscript. Please find below our detailed replies (black font) on your comments (blue font).

My only difficulty with this dataset is that there is a discussion of how uncertainties are calculated, but I cannot find these anywhere in the dataset itself. I had expected each dataset to be matched with an uncertainty estimate. Am I missing something here?  A clarification on this would strengthen the usefulness of the dataset.

The updated version of the dataset includes uncertainties for each provided dataset, except for the global AOD proxy. This version is uploaded, and it can be accessed from the same link.

There are only a couple of minor technical issues that I noticed that am sure will get corrected.  (Line 167 two full stops; highlighting in the "Data availability" section.

Corrected.

**A comment by Alexandre Baron**

This is an interesting dataset that will be of great use for the community. Just a short comment on the eye-catching title. I suggest two additions to be more descriptive of the actual dataset.

1) Adding "Extinction" and maybe "At 750 nm".
2) Adding the time period "1984-2022".

Example: A Climate Data Record of Stratospheric Aerosol Extinctions at 750 nm from 1984 to 2022

**Reply:**

Dear Alexandre,

Thank you for the positive comment on our paper and suggestions for a slight change of the title. However, we prefer to keep the original title. The details about the dataset are

provided in the abstract and in the paper, and the original title is consistent with the name of the dataset, CREST.   We also prefer not to specify a time period in the title, because we are planning to extend the dataset regularly in the future.